# ENABLING FACTORIZED PIANO MUSIC MODELING AND GENERATION WITH THE MAESTRO DATASET

**Curtis Hawthorne⋆, Andriy Stasyuk⋆, Adam Roberts⋆, Ian Simon⋆,**
**Cheng-Zhi Anna Huang⋆, Sander Dieleman†, Erich Elsen⋆, Jesse Engel⋆ & Douglas Eck⋆**
⋆Google Brain, †DeepMind
{fjord,astas,adarob,iansimon,annahuang,sedielem,eriche,jesseengel,deck}@google.com

## ABSTRACT

Generating musical audio directly with neural networks is notoriously difficult because it requires coherently modeling structure at many different timescales. Fortunately, most music is also highly structured and can be represented as discrete note events played on musical instruments. Herein, we show that by using notes as an intermediate representation, we can train a suite of models capable of transcribing, composing, and synthesizing audio waveforms with coherent musical structure on timescales spanning six orders of magnitude (∼0.1 ms to ∼100 s), a process we call Wave2Midi2Wave. This large advance in the state of the art is enabled by our release of the new MAESTRO (MIDI and Audio Edited for Synchronous TRacks and Organization) dataset, composed of over 172 hours of virtuosic piano performances captured with fine alignment (≈3 ms) between note labels and audio waveforms. The networks and the dataset together present a promising approach toward creating new expressive and interpretable neural models of music.

## 1 INTRODUCTION

Since the beginning of the recent wave of deep learning research, there have been many attempts to create generative models of expressive musical audio *de novo*. These models would ideally generate audio that is both musically and sonically realistic to the point of being indistinguishable to a listener from music composed and performed by humans.

However, modeling music has proven extremely difficult due to dependencies across the wide range of timescales that give rise to the characteristics of pitch and timbre (short-term) as well as those of rhythm (medium-term) and song structure (long-term). On the other hand, much of music has a large hierarchy of discrete structure embedded in its generative process: a composer creates songs, sections, and notes, and a performer realizes those notes with discrete events on their instrument, creating sound. The division between notes and sound is in many ways analogous to the division between symbolic language and utterances in speech.

The WaveNet model by van den Oord et al. (2016) may be the first breakthrough in generating musical audio directly with a neural network. Using an autoregressive architecture, the authors trained a model on audio from piano performances that could then generate new piano audio sample-by-sample. However, as opposed to their highly convincing speech examples, which were conditioned on linguistic features, the authors lacked a conditioning signal for their piano model. The result was audio that sounded very realistic at very short time scales (1 or 2 seconds), but that veered off into chaos beyond that.

Dieleman et al. (2018) made great strides towards providing longer term structure to WaveNet synthesis by implicitly modeling the discrete musical structure described above. This was achieved by training a hierarchy of VQ-VAE models at multiple time-scales, ending with a WaveNet decoder to generate piano audio as waveforms. While the results are impressive in their ability to capture long-term structure directly from audio waveforms, the resulting sound suffers from various artifacts at the fine-scale not present in the unconditional WaveNet, clearly distinguishing it from real musical audio. Also, while the model learns a version of discrete structure from the audio, it is not

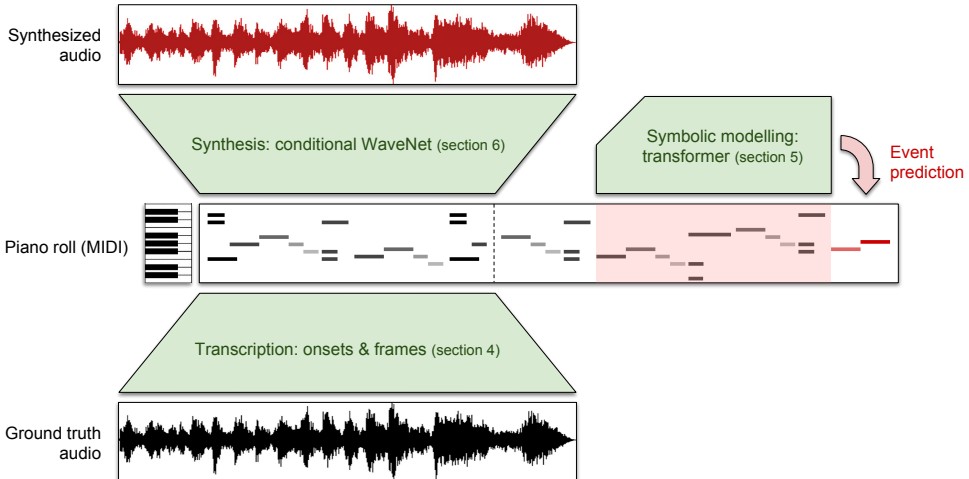

Figure 1: Wave2Midi2Wave system architecture for our suite of piano music models, consisting of (a) a conditional WaveNet model that generates audio from MIDI, (b) a Music Transformer language model that generates piano performance MIDI autoregressively, and (c) a piano transcription model that "encodes" piano performance audio as MIDI.

directly reflective of the underlying generative process and thus not interpretable or manipulable by a musician or user.

Manzelli et al. (2018) propose a model that uses a WaveNet to generate solo cello music conditioned on MIDI notation. This overcomes the inability to manipulate the generated sequence. However, their model requires a large training corpus of labeled audio because they do not train a transcription model, and it is limited to monophonic sequences.

In this work, we seek to explicitly factorize the problem informed by our prior understanding of the generative process of performer and instrument:

$$P(audio) = \mathbb{E}_{notes}\big[P(audio|notes)\big] \tag{1}$$

which can be thought of as a generative model with a discrete latent code of musical notes. Since the latent representation is discrete, and the scale of the problem is too large to jointly train, we split the model into three separately trained modules that are each state-of-the-art in their respective domains:

1. *Encoder, $P(notes|audio)$*: An Onsets and Frames (Hawthorne et al., 2018) transcription model to produce a symbolic representation (MIDI) from raw audio.

2. *Prior, $P(notes)$*: A self-attention-based music language model (Huang et al., 2018) to generate new performances in MIDI format based on those transcribed in (1).

3. *Decoder, $P(audio|notes)$*: A WaveNet (van den Oord et al., 2016) synthesis model to generate audio of the performances conditioned on MIDI generated in (2).

We call this process Wave2Midi2Wave.

One hindrance to training such a stack of models is the lack of large-scale annotated datasets like those that exist for images. We overcome this barrier by curating and publicly releasing alongside this work a piano performance dataset containing well-aligned audio and symbolic performances an order of magnitude larger than the previous benchmarks.

In addition to the high quality of the samples our method produces (see https://goo.gl/magenta/maestro-examples), training a suite of models according to the natural musician/instrument division has a number of other advantages. First, the intermediate representation

used is more suitable for human interpretation and manipulation. Similarly, factorizing the model in this way provides better modularity: it is easy to independently swap out different performance and instrument models. Using an explicit performance representation with modern language models also allows us to model structure at much larger time scales, up to a minute or so of music. Finally, we can take advantage of the large amount of prior work in the areas of symbolic music generation and conditional audio generation. And by using a state-of-the-art music transcription model, we can make use of the same wealth of unlabeled audio recordings previously only usable for training end-to-end models by transcribing unlabeled audio recordings and feeding them into the rest of our model.

## 2    CONTRIBUTIONS OF THIS PAPER

Our contributions are as follows:

1. We combine a transcription model, a language model, and a MIDI-conditioned WaveNet model to produce a factorized approach to musical audio modeling capable of generating about one minute of coherent piano music.

2. We provide a new dataset of piano performance recordings and aligned MIDI, an order of magnitude larger than previous datasets.

3. Using an existing transcription model architecture trained on our new dataset, we achieve state-of-the-art results on a piano transcription benchmark.

## 3    DATASET

We partnered with organizers of the International Piano-e-Competition[1] for the raw data used in this dataset. During each installment of the competition, virtuoso pianists perform on Yamaha Disklaviers which, in addition to being concert-quality acoustic grand pianos, utilize an integrated high-precision MIDI capture and playback system. Recorded MIDI data is of sufficient fidelity to allow the audition stage of the competition to be judged remotely by listening to contestant performances reproduced over the wire on another Disklavier instrument.

The dataset introduced in this paper, which we name MAESTRO ("MIDI and Audio Edited for Synchronous TRacks and Organization"), contains over a week of paired audio and MIDI recordings from nine years of International Piano-e-Competition events.[2] The MIDI data includes key strike velocities and sustain pedal positions. Audio and MIDI files are aligned with ≈3 ms accuracy and sliced to individual musical pieces, which are annotated with composer, title, and year of performance. Uncompressed audio is of CD quality or higher (44.1–48 kHz 16-bit PCM stereo). A train/validation/test split configuration is also proposed, so that the same composition, even if performed by multiple contestants, does not appear in multiple subsets. Repertoire is mostly classical, including composers from the 17th to early 20th century. Table 1 contains aggregate statistics of the MAESTRO dataset.

| Split | Performances | Compositions (approx.) | Duration, hours | Size, GB | Notes, millions |
|---|---|---|---|---|---|
| Train | 954 | 295 | 140.1 | 83.6 | 5.06 |
| Validation | 105 | 60 | 15.3 | 9.1 | 0.54 |
| Test | 125 | 75 | 16.9 | 10.1 | 0.57 |
| **Total** | **1184** | **430** | **172.3** | **102.8** | **6.18** |

Table 1: Statistics of the MAESTRO dataset.

We make the new dataset (MIDI, audio, metadata, and train/validation/test split configuration) available at `https://g.co/magenta/maestro-dataset` under a Creative Commons Attribution Non-Commercial Share-Alike 4.0 license.

---

[1]`http://piano-e-competition.com`
[2]All results in this paper are from the v1.0.0 release of the dataset.

Several datasets of paired piano audio and MIDI have been published previously and have enabled significant advances in automatic piano transcription and related topics. We are making MAE-STRO available because we believe it provides several advantages over existing datasets. Most significantly, as evident from table 2, MAESTRO is around an order of magnitude larger. Existing datasets also have different properties than MAESTRO that affect model training:

**MusicNet** (Thickstun et al., 2017) contains recordings of human performances, but separately-sourced scores. As discussed in Hawthorne et al. (2018), the alignment between audio and score is not fully accurate. One advantage of MusicNet is that it contains instruments other than piano (not counted in table 2) and a wider variety of recording environments.

**MAPS** (Emiya et al., 2010) contains Disklavier recordings and synthesized audio created from MIDI files that were originally entered via sequencer. As such, the "performances" are not as natural as the MAESTRO performances captured from live performances. In addition, synthesized audio makes up a large fraction of the MAPS dataset. MAPS also contains syntheses and recordings of individual notes and chords, not counted in table 2.

**Saarland Music Data (SMD)** (Müller et al., 2011) is similar to MAESTRO in that it contains recordings and aligned MIDI of human performances on a Disklavier, but is 30 times smaller.

| Dataset | Performances | Compositions | Duration, hours | Notes, millions |
|---|---|---|---|---|
| SMD | 50 | 50 | 4.7 | 0.15 |
| MusicNet | 156 | 60 | 15.3 | 0.58 |
| MAPS | 270 | 208 | 17.9 | 0.62 |
| MAESTRO | 1184 | ~430 | 172.3 | 6.18 |

Table 2: Comparison with other datasets.

### 3.1 ALIGNMENT

Our goal in processing the data from International Piano-e-Competition was to produce pairs of audio and MIDI files time-aligned to represent the same musical events. The data we received from the organizers was a combination of MIDI files recorded by Disklaviers themselves and WAV audio captured with conventional recording equipment. However, because the recording streams were independent, they differed widely in start times and durations, and they were also subject to jitter. Due to the large volume of content, we developed an automated process for aligning, slicing, and time-warping provided audio and MIDI to ensure a precise match between the two.

Our approach is based on globally minimizing the distance between CQT frames from the real audio and synthesized MIDI (using *FluidSynth*[3]). Obtaining a highly accurate alignment is non-trivial, and we provide full details in the appendix.

### 3.2 DATASET SPLITTING

For all experiments in this paper, we use a single train/validation/test split designed to satisfy the following criteria:

- No composition should appear in more than one split.
- Train/validation/test should make up roughly 80/10/10 percent of the dataset (in time), respectively. These proportions should be true globally and also within each composer. Maintaining these proportions is not always possible because some composers have too few compositions in the dataset.
- The validation and test splits should contain a variety of compositions. Extremely popular compositions performed by many performers should be placed in the training split.

For comparison with our results, we recommend using the splits which we have provided. We do not necessarily expect these splits to be suitable for all purposes; future researchers are free to use alternate experimental methodologies.

___
[3]http://www.fluidsynth.org/

## 4    PIANO TRANSCRIPTION

The large MAESTRO dataset enables training an automatic piano music transcription model that achieves a new state of the art. We base our model on Onsets and Frames, with several modifications informed by a coarse hyperparameter search using the validation split. For full details of the base model architecture and training procedure, refer to Hawthorne et al. (2018).

One important modification was adding an offset detection head, inspired by Kelz et al. (2018). The offset head feeds into the frame detector but is not directly used during decoding. The offset labels are defined to be the 32ms following the end of each note.

We also increased the size of the bidirectional LSTM layers from 128 to 256 units, changed the number of filters in the convolutional layers from 32/32/64 to 48/48/96, and increased the units in the fully connected layer from 512 to 768. We also stopped gradient propagation into the onset subnetwork from the frame network, disabled weighted frame loss, and switched to HTK frequency spacing (Young et al., 2006) for the mel-frequency spectrogram input. In general, we found that the best ways to get higher performance with the larger dataset were to make the model larger and simpler.

The final important change we made was to start using audio augmentation during training using an approach similar to the one described in McFee et al. (2017). During training, every input sample was modified using random parameters for the *SoX* [4] audio tool using *pysox* (Bittner et al., 2016). The parameters, ranges, and random sampling methods are described in table 3. We found that this was particularly important when evaluating on the MAPS dataset, likely because the audio augmentation made the model more robust to differences in recording environment and piano qualities. The differences in training results are summarized in table 6. When evaluating against the MAESTRO dataset, we found that audio augmentation made results slightly worse, so results in table 5 are presented without audio augmentation.

| Description | Scale | Range | Sampling |
|---|---|---|---|
| pitch shift | semitones | $-0.1$–$0.1$ | linear |
| contrast (compression) | amount | 0.0–100.0 | linear |
| equalizer 1 | frequency | 32.0–4096.0 | log |
| equalizer 2 | frequency | 32.0–4096.0 | log |
| reverb | reverberance | 0.0–70.0 | log |
| pinknoise | volume | 0.0–0.04 | linear |

Table 3: Audio augmentation parameters.

After training on the MAESTRO training split for 670k steps, we achieved state of the art results described in table 4 for the MAPS dataset. We also present our results on the train, validation, and test splits of the MAESTRO dataset as a new baseline score in table 5. Note that for calculating the scores of the train split, we use the full duration of the files without splitting them into 20-second chunks as is done during training.

| | Frame | | | Note | | | Note w/ offset | | | Note w/ offset & velocity | | |
|---|---|---|---|---|---|---|---|---|---|---|---|---|
| | P | R | F1 | P | R | F1 | P | R | F1 | P | R | F1 |
| Hawthorne et al. (2018) | 88.53 | 70.89 | 78.30 | 84.24 | 80.67 | 82.29 | 51.32 | 49.31 | 50.22 | 35.52 | 30.80 | 35.39 |
| Kelz et al. (2018) | 90.73 | 67.85 | 77.16 | **90.15** | 74.78 | 81.38 | 61.93 | 51.66 | 56.08 | — | — | — |
| Onsets & Frames (MAESTRO) | **92.86** | **78.46** | **84.91** | 87.46 | **85.58** | **86.44** | **68.22** | **66.75** | **67.43** | **52.41** | **51.22** | **51.77** |

Table 4: Transcription Precision, Recall, and F1 Results on MAPS configuration 2 test dataset (ENSTDkCl and ENSTDkAm full-length .wav files). Training was done on the MAESTRO trianing set with audio augmentation. Scores are calculated using the same method as in Hawthorne et al. (2018). Note-based scores calculated by the *mir_eval* library, frame-based scores as defined in Bay et al. (2009). Final metric is the mean of scores calculated per piece.

In sections 5 and 6, we demonstrate how using this transcription model enables training language and synthesis models on a large set of unlabeled piano data. To do this, we transcribe the audio in the MAESTRO training set, although in theory any large set of unlabeled piano music would work. We

---

[4]http://sox.sourceforge.net/

| | Frame | | | Note | | | Note w/ offset | | | Note w/ offset & velocity | | |
|---|---|---|---|---|---|---|---|---|---|---|---|---|
| | P | R | F1 | P | R | F1 | P | R | F1 | P | R | F1 |
| Train | 94.23 | 92.58 | 93.35 | 98.88 | 94.41 | 96.56 | 88.13 | 84.19 | 86.09 | 84.98 | 81.20 | 83.02 |
| Validation | 91.69 | 87.80 | 89.58 | 98.42 | 92.61 | 95.38 | 82.93 | 78.17 | 80.44 | 80.36 | 75.75 | 77.95 |
| Test | 92.11 | 88.41 | 90.15 | 98.27 | 92.61 | 95.32 | 82.95 | 78.24 | 80.50 | 79.89 | 75.37 | 77.54 |

Table 5: Results from training the modified Onsets and Frames model on the MAESTRO train split without audio augmentation. Precision, Recall, and F1 Results on the splits of the MAESTRO dataset. Calculations done in the same manner as table 4.

| | Frame F1 | Note F1 | Note w/ offset F1 | Note w/ offset & velocity F1 |
|---|---|---|---|---|
| With Audio Augmentation (MAPS) | **84.91** | **86.44** | **67.43** | **51.77** |
| Without Audio Augmentation (MAPS) | 82.02 | 83.04 | 61.84 | 48.07 |
| With Audio Augmentation (MAESTRO) | 89.19 | 94.80 | 79.67 | 76.04 |
| Without Audio Augmentation (MAESTRO) | **90.15** | **95.32** | **80.50** | **77.54** |

Table 6: Comparisons between results of training the same model with the same number of steps (670k) with and without audio augmentation. Training was done on the MAESTRO training set and evaluation was on either the MAPS configuration 2 test set as in table 4 or the MAESTRO test set as in table 5.

call this new, transcribed version of the training set MAESTRO-T. While it is true that the audio transcribed for MAESTRO-T was also used to train the transcription model, table 5 shows that the model performance is not significantly different between the training split and the validation or test splits, and we needed the larger split to enable training the other models.

## 5 MUSIC TRANSFORMER TRAINING

For our generative language model, we use the decoder portion of a Transformer (Vaswani et al., 2017) with relative self-attention, which has previously shown compelling results in generating music with longer-term coherence (Huang et al., 2018). We trained two models, one on MIDI data from the MAESTRO dataset and another on MIDI transcriptions inferred by Onsets and Frames from audio in MAESTRO, referred to as MAESTRO-T in section 4. For full details of the model architecture and training procedure, refer to Huang et al. (2018).

We used the same training procedure for both datasets. We trained on random crops of 2048 events and employed transposition and time compression/stretching data augmentation. The transpositions were uniformly sampled in the range of a minor third below and above the original piece. The time stretches were at discrete amounts and uniformly sampled from the set $\{0.95, 0.975, 1.0, 1.025, 1.05\}$.

We evaluated both of the models on their respective validation splits.

| Model variation | NLL on their respective validation splits |
|---|---|
| Music Transformer trained on MAESTRO | 1.84 |
| Music Transformer trained on MAESTRO-T | 1.72 |

Table 7: Validation Negative Log-Likelihood, with event-based representation.

Samples outputs from the Music Transformer model can be heard in the Online Supplement (https://goo.gl/magenta/maestro-examples).

## 6 PIANO SYNTHESIS

Most commercially available systems that are able to synthesize a MIDI sequence into a piano audio signal are concatenative: they stitch together snippets of audio from a large library of recordings of individual notes. While this stitching process can be quite ingenious, it does not optimally capture the various interactions between notes, whether they are played simultaneously or in sequence. An alternative but less popular strategy is to simulate a physical model of the instrument. Constructing an accurate model constitutes a considerable engineering effort and is a field of research by itself (Bank et al., 2010; Valimaki et al., 2012).

WaveNet (van den Oord et al., 2016) is able to synthesize realistic instrument sounds directly in the waveform domain, but it is not as adept at capturing musical structure at timescales of seconds or longer. However, if we provide a MIDI sequence to a WaveNet model as conditioning information, we eliminate the need for capturing large scale structure, and the model can focus on local structure instead, i.e., instrument timbre and local interactions between notes. Conditional WaveNets are also used for text-to-speech (TTS), and have been shown to excel at generating realistic speech signals conditioned on linguistic features extracted from textual data. This indicates that the same setup could work well for music audio synthesis from MIDI sequences.

Our WaveNet model uses a similar autoregressive architecture to van den Oord et al. (2016), but with a larger receptive field: 6 (instead of 3) sequential stacks with 10 residual block layers each. We found that a deeper context stack, namely 2 stacks with 6 layers each arranged in a series, worked better for this task. We also updated the model to produce 16-bit output using a mixture of logistics as described in van den Oord et al. (2018).

The input to the context stack is an onset "piano roll" representation, a size-88 vector signaling the onset of any keys on the keyboard, with 4ms bins (250Hz). Each element of the vector is a float that represents the strike velocity of a piano key in the 4ms frame, scaled to the range [0, 1]. When there is no onset for a key at a given time, the value is 0.

We initially trained three models:

**Unconditioned** Trained only with the audio from the combined MAESTRO training/validation splits with no conditioning signal.

**Ground** Trained with the ground truth audio/MIDI pairs from the combined MAESTRO training/validation splits.

**Transcribed** Trained with ground truth audio and MIDI inferred from the audio using the Onsets and Frames method, referred to as MAESTRO-T in section 4.

The resulting losses after 1M training steps were 3.72, 3.70 and 3.84, respectively. Due to teacher forcing, these numbers do not reflect the quality of conditioning, so we rely on human judgment for evaluation, which we address in the following section.

It is interesting to note that the WaveNet model recreates non-piano subtleties of the recording, including the response of the room, breathing of the player, and shuffling of listeners in their seats. These results are encouraging and indicate that such methods could also capture the sound of more dynamic instruments (such as string and wind instruments) for which convincing synthesis/sampling methods lag behind piano.

Due to the heterogeneity of the ground truth audio quality in terms of microphone placement, ambient noise, etc., we sometime notice "timbral shifts" during longer outputs from these models. We therefore additionally trained a model conditioned on a one-hot year vector at each timestep (similar to speaker conditioning in TTS), which succeeds in producing consistent timbres and ambient qualities during long outputs (see Online Supplement).

A side effect of arbitrary windowing of the training data across note boundaries is a sonic crash that often occurs at the beginning of generated outputs. To sidestep this issue, we simply trim the first 2 seconds of all model outputs reported in this paper, and in the Online Supplement (`https://goo.gl/magenta/maestro-examples`).

## 7    LISTENING TESTS

Since our ultimate goal is to create realistic musical audio, we carried out a listening study to determine the perceived quality of our method. To separately assess the effects of transcription, language modeling, and synthesis on the listeners' responses, we presented users with two 20-second clips[5] drawn from the following sets, each relying on an additional model from our factorization:

**Ground Truth Recordings** Clips randomly selected from the MAESTRO validation audio split.

---

[5]While it would be beneficial to do a listening study on longer samples, running a listening study on those samples at scale was not feasible.

**WaveNet Unconditioned** Clips generated by the Unconditioned WaveNet model described in section 6.

**WaveNet Ground/Test** Clips generated by the Ground WaveNet model described in section 6, conditioned on random 20-second MIDI subsequences from the MAESTRO test split.

**WaveNet Transcribed/Test** Clips generated by the Transcribed WaveNet model described in section 6, conditioned on random 20-second subsequences from the MAESTRO test split.

**WaveNet Transcribed/Transformer** Clips generated by the Transcribed WaveNet model described in section 6, conditioned on random 20-second subsequences from the Music Transformer model described in section 5 that was trained on MAESTRO-T.

The final set of samples demonstrates the full end-to-end ability of taking unlabeled piano performances, inferring MIDI labels via transcription, generating new performances with a language model trained on the inferred MIDI, and rendering new audio as though it were played on a similar piano—all without any information other than raw audio recordings of piano performances.

Participants were asked which clip they thought sounded more like a recording of somebody playing a musical piece on a real piano, on a Likert scale. 640 ratings were collected, with each source involved in 128 pair-wise comparisons. Figure 2 shows the number of comparisons in which performances from each source were selected as more realistic.

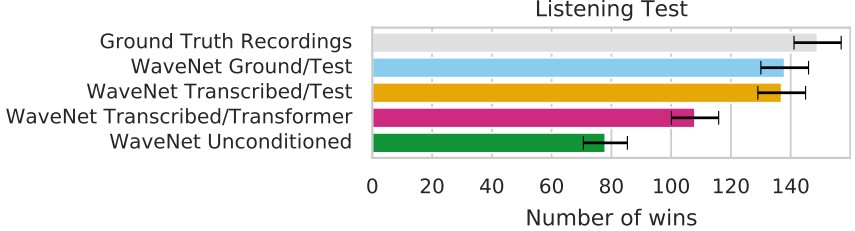

Figure 2: Results of our listening tests, showing the number of times each source won in a pairwise comparison. Black error bars indicate estimated standard deviation of means.

A Kruskal-Wallis H test of the ratings showed that there is at least one statistically significant difference between the models: $\chi^2(2) = 67.63, p < 0.001$. A post-hoc analysis using the Wilcoxon signed-rank test with Bonferroni correction showed that there was not a statistically significant difference in participant ratings between real recordings and samples from the WaveNet Ground/Test and WaveNet Transcribed/Test models with $p > 0.01/10$.

Audio of some of the examples used in the listening tests is available in the Online Supplement (`https://goo.gl/magenta/maestro-examples`).

## 8 CONCLUSION

We have demonstrated the Wave2Midi2Wave system of models for factorized piano music modeling, all enabled by the new MAESTRO dataset. In this paper we have demonstrated all capabilities on the same dataset, but thanks to the new state-of-the-art piano transcription capabilities, any large set of piano recordings could be used,[6] which we plan to do in future work. After transcribing the recordings, the transcriptions could be used to train a WaveNet and a Music Transformer model, and then new compositions could be generated with the Transformer and rendered with the WaveNet. These new compositions would have similar musical characteristics to the music in the original dataset, and the audio renderings would have similar acoustical characteristics to the source piano.

The most promising future work would be to extend this approach to other instruments or even multiple simultaneous instruments. Finding a suitable training dataset and achieving sufficient transcription performance will likely be the limiting factors.

---

[6]The performance of the transcription model on the separate MAPS dataset in table 4 shows that the model effectively generalizes beyond just the MAESTRO recordings.

The new dataset (MIDI, audio, metadata, and train/validation/test split configurations) is available at `https://g.co/magenta/maestro-dataset` under a Creative Commons Attribution Non-Commercial Share-Alike 4.0 license. The Online Supplement, including audio examples, is available at `https://goo.gl/magenta/maestro-examples`.

ACKNOWLEDGMENTS

We would like to thank Michael E. Jones and Stella Sick for their help in coordinating the release of the source data and Colin Raffel for his careful review and comments on this paper.

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

APPENDIX

In this appendix, we describe in detail how the MAESTRO dataset from section 3 was aligned and segmented.

COARSE ALIGNMENT

The key idea for the alignment process was that even an untrained human can recognize whether two performances are of the same score based on raw audio, disregarding differences in the instrument or recording equipment used. Hence, we synthesized the provided MIDI (using *FluidSynth* with a SoundFont sampled from recordings of a Disklavier[7]) and sought to define an audio-based difference metric that could be minimized to find the best-alignment shift for every audio/MIDI pair.

We wanted the metric to take harmonic features into account, so as a first step we used *librosa* (McFee et al., 2017) to compute the Constant-Q Transform (Brown, 1991; Schörkhuber & Klapuri, 2010) of both original and synthesized audio. For the initial alignment stage we picked a hop length of 4096 samples (∼90 ms) as a trade-off between speed and accuracy, which proved reasonable for most of the repertoire.[8] Microphone setup varied between competition years and stages, resulting in varying frequency response and overall amplitude levels in recordings, especially in the lower and higher ends of the piano range. To account for that, we limited the CQT to 48 buckets aligned with MIDI notes C2–B5, and also converted amplitude levels to dB scale with maximum absolute amplitude as a reference point and a hard cut-off at -80 dB. Original and synthesized audio also differed in sound decay rate, so we normalized the resulting CQT arrays time-wise by dividing each hop column by its minimum value (averaged over a 5-hop window).

A single MIDI file from a Disklavier typically covered several hours of material corresponding to a sequence of shorter audio files from several seconds up to an hour long. We slid the normalized CQT of each such original audio file against a window of synthesized MIDI CQT of the same length and used mean squared error (MSE) between the two as the difference metric.[9] Minimum error determined best alignment, after which we attempted to align the next audio file in sequence with the remaining length of the corresponding MIDI file. Due to the length of MIDI files, it was impractical to calculate MSE at each possible shift, so instead we trimmed silence at the beginning of audio, and attempted to align it with the first "note on" event of the MIDI file, within ±12 minutes tolerance. If the minimum error was still high, we attempted alignment at the next "note on" event after a 30-second silence.

This approach allowed us to skip over unusable sections of MIDI recordings that did not correspond to audio, e.g., instrument tuning and warm-ups, and also non-musical segments of audio such as applause and announcements. Non-piano sounds also considerably increased the MSE metric for very short audio files, so we had to either concatenate those with their longer neighbors if they had any musical material or exclude them completely. Events that were present at the beginning of audio files beyond the chosen shift tolerance which did not correspond to MIDI had to be cut off manually. In order to recover all musically useful data we also had to manually repair several MIDI files where the clock had erroneously jumped, causing the remainder of the file to be out of sync with corresponding audio.

After tuning process parameters and addressing the misaligned audio/MIDI pairs detected by unusually high CQT MSE, we have reached the state where each competition year (i.e., different audio recording setup) has final metric values for all pairs within a close range. Spot-checking the pairs with the highest MSE values for each year confirmed proper alignment, which allowed us to proceed to the segmentation stage.

---

[7] http://freepats.zenvoid.org/sf2/acoustic_grand_piano_ydp_20080910.txt

[8] Notably, Rimsky-Korsakov's "Flight of the Bumblebee", known for its rapid tempo, approached the limit of chosen hop length, yielding slightly higher difference metric values even at best alignment.

[9] Testing this metric empirically on each competition year for an aligned audio/MIDI pair versus two completely different audio segments showed ∼2–2.5x difference in metric values, which provided sufficient separation for our purpose.

SEGMENTATION

Since certain compositions were performed by multiple contestants,[10] we needed to segment the aligned audio/MIDI pairs further into individual musical pieces, so as to enable splitting the data into train, validation, and test sets disjoint on compositions. While the organizers provided the list of composition metadata for each audio file, for some competition years timing information was missing. In such cases we greedily sliced audio/MIDI pairs at the longest silences between MIDI notes up to the expected number of musical pieces.

Where expected piece duration data was available, we applied search with backtracking roughly as follows. As an invariant, the segmentation algorithm maintained a list of intervals as start–end time offsets along with a list of expected piece durations, so that the total length of the piece durations corresponding to each interval was less than the interval duration (within a certain tolerance). At each step we picked the next longest MIDI silence and determined which interval it belonged to. Then we split that interval in two at the silence and attempted to split the corresponding sequence of durations as well, satisfying the invariant. For each suitable split the algorithm continued to the next longest silence. If multiple splits were possible, the algorithm preferred the ones that divided the piece durations more evenly according to a heuristic. If no such split was possible, the algorithm either skipped current silence if it was short[11] and attempted to split at the next one, or backtracked otherwise. It also backtracked if no more silences longer than 3 seconds were available. The algorithm succeeded as soon as each interval corresponded to exactly one expected piece duration.

Once a suitable segmentation was found, we sliced each audio/MIDI pair at resulting intervals, additionally trimming short clusters of notes at the beginning or end of each segment that appeared next to long MIDI silences in order to cut off additional non-music events (e.g., tuning or contestants testing the instrument during applause), and adding an extra 1 second of padding at both ends before making the final cut.

FINE ALIGNMENT

After the initial alignment and segmentation, we applied Dynamic Time Warping (DTW) to account for any jitter in either the audio or MIDI recordings. DTW has seen wide use in audio-to-MIDI alignment; for an overview see Müller (2015). We follow the *align_midi* example from *pretty_midi* (Raffel & Ellis, 2014), except that we use a custom C++ DTW implementation for improved speed and memory efficiency to allow for aligning long sequences.

First, in Python, we use *librosa* to load the audio and resample it to a 22,050Hz mono signal. Next, we load the MIDI and synthesize it at the same sample rate, using the same *FluidSynth* process as above. Then, we pad the end of the shorter of the two sample arrays so they are the same length. We use the same procedure as *align_midi* to extract CQTs from both sample arrays, except that we use a hop length of 64 to achieve a resolution of $\sim$3ms. We then pass these CQTs to our C++ DTW implementation. To avoid calculating the full distance matrix and taking its mean to get a penalty value, we instead sample 100k random pairs and use the mean of their cosine distances.

We use the same DTW algorithm as implemented in *librosa* except that we calculate cosine distances only within a Sakoe-Chiba band radius (Sakoe & Chiba, 1978) of 2.5 seconds instead of calculating distances for all pairs. Staying within this small band limits the number of calculations we need to make and the number of distances we have to store in memory. This is possible because we know from the previous alignment pass that the sequences are already mostly aligned and we just need to account for small constant offsets due to the lower resolution of the previous process and apply small sequence warps to recover from any occasional jitter.

---

[10]Some competition stages have a specific list of musical pieces or composers to choose from.

[11]The reasoning being that long silences must appear between different compositions in the performance, whereas shorter ones might separate individual movements of a single composition, which could belong to a single audio/MIDI pair after segmentation.

