# OpenReview forum: "Enabling Factorized Piano Music Modeling and Generation with the MAESTRO Dataset"
_ICLR.cc/2019/Conference_

### Official Review · AnonReviewer2 · 2018-11-03
**Put three state of the art models together and get impressive results for modeling piano music.**

**Rating:** 8
**Confidence:** 4

**Review:**

This paper combines state of the art models for piano transcription, symbolic music synthesis, and waveform generation all using a shared piano-roll representation.  It also introduces a new dataset of 172 hours of aligned MIDI and audio from real performances recorded on Yamaha Disklavier pianos in the context of the piano-e-competition.

By using this shared representation and this dataset, it is able to expand the amount of time that it can coherently model music from a few seconds to a minute, necessary for truly modeling entire musical pieces.

Training an existing state of the art transcription model on this data improves performance on a standard benchmark by several percentage points (depending on the specific metric used).

Listening test results show that people still prefer the real recordings a plurality of the time, but that the syntheses are selected over them a fair amount.  One thing that is clear from the audio examples is that the different systems produce output with different equalization levels, which may lead to some of the listening results.  If some sort of automatic mastering were done to the outputs this might be avoided.

While the novelty of the individual algorithms is relatively meager, their combination is very synergistic and makes a significant contribution to the field.  Piano music modeling is a long-standing problem that the current paper has made significant progress towards solving.

The paper is very well written, but there are a few minor issues:
* Eq (1) this is really the joint distribution between audio and notes, not the marginal of audio
* Table 4: What do precision, recall, and f1 score mean for notes with velocity?  How close does the system have to be to the velocity to get it right?
* Table 6: NLL presumably stands for Negative Log Likelihood, but this should be made explicity
* Figure 2: Are the error bars the standard deviation of the mean or the standard error of the mean?

---

> ### Author Response · Authors · 2018-11-14
> **Response for AnonReviewer2**
>
> Thank you for your review and comments.
>
> * Eq (1) this is really the joint distribution between audio and notes, not the marginal of audio
>
> Thank you for catching the mistake. We have updated the equation to include the marginalizing integral through the expectation over notes: P(audio) = E_{notes} [ P(audio|notes) ]
>
> * Table 4: What do precision, recall, and f1 score mean for notes with velocity?  How close does the system have to be to the velocity to get it right?
>
> We use the mir_eval library for calculating those metrics, and a full description is available here: https://craffel.github.io/mir_eval/#module-mir_eval.transcription_velocity
>
> It implements the evaluation procedure described in Hawthorne et al. (2018).
>
> We have updated the caption for Table 4 to make this more clear.
>
> * Table 6: NLL presumably stands for Negative Log Likelihood, but this should be made explicitly
>
> Thanks, updated the table caption to make this more clear.
>
> * Figure 2: Are the error bars the standard deviation of the mean or the standard error of the mean?
>
> We are calculating the standard deviation of the means (we did not divide by the square root of the sample size).

---

### Official Review · AnonReviewer1 · 2018-11-07
**Learning to generate piano music via MIDI layer**

**Rating:** 8
**Confidence:** 2

**Review:**

The paper addresses the challenge of using neural networks to generate original and expressive piano music.  The available techniques today for audio or music generation are not able to sufficient handle the many levels at which music needs to modeled.  The result is that while individual music sounds (or notes) can be generated at one level using tools like WaveNet, they don't come together to create a coherent work of music at the higher level.  The paper proposes to address this problem by imposing a MIDI representation (piano roll) in the neural modeling of music audio that serves as an intermediate (and interpretable) representation between the analysis (music audio -> MIDI) and synthesis (MIDI -> music audio) in the pipeline of piano music generation.  In order to develop and validate the proposed learning architecture, the authors have created a large data set of aligned piano music (raw audio along with MIDI representation).  Using this data set for training, validation and test, the paper reports on listening tests that showed slightly less favorable results for the generated music.  A few questions and comments are as follows.  MIDI itself is a rich language with ability to drive the generation of music using rich sets of customizable sound fonts.  Given this, it is not clear that it is necessary to reproduce this function using neural network generation of sounds.  The further limitation of the proposed approach seems to be the challenge of decoding raw music audio with chords, multiple overlayed notes or multiple tracks.  MIDI as a representation can support multiple tracks, so it is not necessarily the bottleneck.  How much does the data augmentation (audio augmentation) help?

---

> ### Author Response · Authors · 2018-11-14
> **Response for AnonReviewer1**
>
> Thank you for your review and comments.
>
> * MIDI itself is a rich language with ability to drive the generation of music using rich sets of customizable sound fonts.  Given this, it is not clear that it is necessary to reproduce this function using neural network generation of sounds.
>
> Synthesizing realistic audio from symbolic representations is a complex task. While there are many good sounding piano synthesizers, many of them fall well short of producing audio that would be a convincing substitute for a real piano recording. For example, the SoundFont technology referenced can only play particular samples for particular notes (with some simple effects processing). It is incapable of modeling complex physical interactions between different parts of the piano, such as sympathetic resonance, and is limited by the quality and variety of samples included with a particular font (for example, the ability to play longer notes is often achieved by simply looping over a section of a sample). That said, there are some piano synthesis systems that can do a good job of modeling these types of interactions, though they are not as widely available as SoundFonts and are difficult to create. For a good overview of the difficulties and successes in piano modeling, see the paper we cited by Bank et al.
>
> Our WaveNet model is able to learn to generate realistic-sounding music with no information other than audio recordings of piano performances, information which would be insufficient for the creation of a SoundFont or physics-informed model. The “Transcribed” WaveNet model clearly demonstrates this because we use only the audio from the dataset and we derive training labels by using our transcription model. By training on the audio directly, we implicitly model the complex physical interactions of the instrument, unlike a SoundFont.
>
> It is also interesting to note that the WaveNet model recreates non-piano subtleties of the recording, including the response of the room, breathing of the player, and shuffling of listeners in their seats. These results are encouraging and indicate that such methods could also capture the sound of more dynamic instruments (such as string and wind instruments) for which convincing synthesis/sampling methods lag behind piano. To clarify this point, we have added a paragraph to the Piano Synthesis section of the paper.
>
> We have also updated the paper to further demonstrate our ability to control the output sound by adding year conditioning. Different competition years within the MAESTRO dataset had different microphone placements (e.g., near the piano or farther back in the room), and by conditioning on year, we can control whether the output sounds like a close mic recording or one with more room noise. We present several audio examples in the online supplement: https://goo.gl/6RzHZM
>
> * The further limitation of the proposed approach seems to be the challenge of decoding raw music audio with chords, multiple overlaid notes or multiple tracks.  MIDI as a representation can support multiple tracks, so it is not necessarily the bottleneck.
>
> We chose to model the music with full polyphony for a couple reasons. One is that, as described above, there are complex interactions in the physical piano and recording environment that would not be reproducible by rending notes separately and then layering them into a single output. Another is that the training data is presented as a single MIDI stream and the audio is not easily separated into multiple tracks.
>
> * How much does the data augmentation (audio augmentation) help?
>
> We have added a table showing the differences between training with and without audio augmentation. In the process of analyzing these results, we realized that audio augmentation helps significantly when evaluating on the MAPS dataset (likely because the model is more robust to differences in recording environment and piano qualities), it actually incurs a slight penalty when evaluating on the MAESTRO test set. We have updated the paper with a discussion of these differences.

---

### Official Review · AnonReviewer3 · 2018-11-10
**Large dataset of parallel MIDI/Audio enables better piano music transcription, synthesis, and generation**

**Rating:** 8
**Confidence:** 5

**Review:**

This paper describes a new large scale dataset of aligned MIDI and audio from real piano performances and presents experiments using several existing state-of-the-art models for transcription, synthesis, and generation. As a result of the new dataset being nearly an order of magnitude larger than existing resources, each component model (with some additional tuning to increase capacity) yields impressive results, outperforming the current state-of-the-art on each component task.
Overall, while the modeling advances here are small if any, I think this paper represents a solid case study in collecting valuble supervised data to push a set of tasks forward. The engineering is carefully done, well-motivated, and clearly described. The results are impressive on all three tasks. Finally, if the modeling ideas here do not, the dataset itself will go on to influence and support this sub-field for years to come.
Comments / questions:
-Is MAPS actually all produced via sequencer? Having worked with this data I can almost swear that at least a portion of it (in particular, the data used here for test) sounds like live piano performance captured on Disklavier. Possibly I'm mistaken, but this is worth a double check.
-Refering to the triple of models as an auto-encoder makes me slightly uncomfortable given that they are all trained independently, directly from supervised data.
-The MAESTRO-T results are less interesting than they might appear at first glance given that the transcriptions are from train. The authors do clearly acknowledge this, pointing out that val and test transcription accuracies were near train accuracy. But maybe that same argument could be used to support that the pure MAESTRO results are themselves generalizable, allowing the authors to simplify slightly by removing MAESTRO-T altogether. In short, I'm not sure MAESTRO-T results offer much over MAESTRO results, and could therefore could be omitted.

---

> ### Author Response · Authors · 2018-11-14
> **Response for AnonReviewer3**
>
> Thank you for your review and comments.
>
> * Is MAPS actually all produced via sequencer? Having worked with this data I can almost swear that at least a portion of it (in particular, the data used here for test) sounds like live piano performance captured on Disklavier. Possibly I'm mistaken, but this is worth a double check.
>
> According to the PDF file that accompanies the MAPS dataset (“MAPS - A piano database for multipitch estimation and automatic transcription of music”): “These high quality files have been carefully hand-written in order to obtain a kind of musical interpretation as a MIDI file.” We have updated the citation to point to this paper specifically to make things more clear. More information about the process is available on the website that contains the source MIDI files for MAPS: http://www.piano-midi.de/technic.htm
>
> * Referring to the triple of models as an auto-encoder makes me slightly uncomfortable given that they are all trained independently, directly from supervised data.
>
> This is a very reasonable point, because there are no learned feature vectors in the latent representation (they come from labels). We have updated the text to instead refer to the model as a “generative model with a discrete latent code of musical notes”. We have kept the encoder/decoder/prior notation because it still seems appropriate.
>
> * The MAESTRO-T results are less interesting than they might appear at first glance given that the transcriptions are from train. The authors do clearly acknowledge this, pointing out that val and test transcription accuracies were near train accuracy. But maybe that same argument could be used to support that the pure MAESTRO results are themselves generalizable, allowing the authors to simplify slightly by removing MAESTRO-T altogether. In short, I'm not sure MAESTRO-T results offer much over MAESTRO results, and could therefore could be omitted.
>
> Our goal with the MAESTRO-T dataset was to clearly demonstrate that both the language modeling tasks (Music Transformer) and audio synthesis (WaveNet) can produce compelling results without having access to ground truth labels. We agree that using the train dataset does somewhat diminish this demonstration, but argue that it does more clearly demonstrate the usefulness of the “Wave2Midi2Wave” process than just using ground truth labels. In future work, we plan to expand our use of these models to datasets that do not have ground truth labels. We have added to the conclusion to clarify this point.

---

### Author Response · Authors · 2018-11-14
**Update**

Thank you to all reviewers for your careful review and comments on the paper. We will address specific questions in responses to particular reviews, but we also wanted to highlight some general updates we have made since the initial submission of the paper:

Our transcription results have improved (Note w/ offset F1 score on MAPS configuration 2 test went from 64.03 to 66.33) due to two modifications:
* We added an offset detection head to the model, inspired by Kelz et al. (2018).
* We trained the transcription for more steps (670k instead of 178k).

Our synthesis results have improved because we switched to using a larger receptive field for the Piano Synthesis WaveNet model (6 instead of 3 sequential stacks).

In order to more accurately compare our WaveNet models, we also trained an unconditioned WaveNet model trained only with the audio from the combined MAESTRO training/validation splits with no conditioning signal.

We improved our listening study by:
* Rerunning it with the improved WaveNet model
* Switching to 20-second samples instead of 10-second samples
* Clarifying our question to ask the raters which clip they thought sounded more like a recording of somebody playing a musical piece on a real piano.

The study results now show that there is not a statistically significant difference in participant ratings between real recordings and samples from the WaveNet Ground/Test and WaveNet Transcribed/Test models.

To better control the timbre of synthesis output, we implemented year conditioning, which can produce outputs that mimic the microphone placement of the different competition years in the dataset.

Finally, we decided to name the process of transcription, MIDI manipulation, and then synthesis Wave2Midi2Wave.

---

### Meta-Review · Area_Chair1 · 2018-12-20

**Confidence:** 5
**Recommendation:** Accept (Oral)

**Metareview:**

All reviewers agree that the presented audio data augmentation is very interesting, well presented, and clearly advancing the state of the art in the field. The authors’ rebuttal clarified the remaining questions by the reviewers. All reviewers recommend strong acceptance (oral presentation) at ICLR. I would like to recommend this paper for oral presentation due to a number of reasons including the importance of the problem addressed (data augmentation is the only way forward in cases where we do not have enough of training data), the novelty and innovativeness of the model, and the clarity of the paper. The work will be of interest to the widest audience beyond ICLR.